# Efficiently escaping saddle points on manifolds

**Chris Criscitiello**
Department of Mathematics
Princeton University
Princeton, NJ 08544
ccriscitiello6@gmail.com

**Nicolas Boumal**
Department of Mathematics
Princeton University
Princeton, NJ 08544
nboumal@math.princeton.edu

## Abstract

Smooth, non-convex optimization problems on Riemannian manifolds occur in machine learning as a result of orthonormality, rank or positivity constraints. First- and second-order necessary optimality conditions state that the Riemannian gradient must be zero, and the Riemannian Hessian must be positive semidefinite. Generalizing Jin et al.'s recent work on perturbed gradient descent (PGD) for optimization on linear spaces [How to Escape Saddle Points Efficiently (2017) [17], Stochastic Gradient Descent Escapes Saddle Points Efficiently (2019) [18]], we propose a version of perturbed Riemannian gradient descent (PRGD) to show that necessary optimality conditions can be met approximately with high probability, without evaluating the Hessian. Specifically, for an arbitrary Riemannian manifold $\mathcal{M}$ of dimension $d$, a sufficiently smooth (possibly non-convex) objective function $f$, and under weak conditions on the retraction chosen to move on the manifold, with high probability, our version of PRGD produces a point with gradient smaller than $\epsilon$ and Hessian within $\sqrt{\epsilon}$ of being positive semidefinite in $O((\log d)^4/\epsilon^2)$ gradient queries. This matches the complexity of PGD in the Euclidean case. Crucially, the dependence on dimension is low. This matters for large-scale applications including PCA and low-rank matrix completion, which both admit natural formulations on manifolds. The key technical idea is to generalize PRGD with a distinction between two types of gradient steps: "steps on the manifold" and "perturbed steps in a tangent space of the manifold." Ultimately, this distinction makes it possible to extend Jin et al.'s analysis seamlessly.

## 1 Introduction

Machine learning has stimulated interest in obtaining global convergence rates in non-convex optimization. Consider a possibly non-convex objective function $f\colon \mathbb{R}^d \to \mathbb{R}$. We want to solve

$$\min_{x \in \mathbb{R}^d} f(x). \tag{1}$$

This is hard in general. Instead, we usually settle for approximate first-order critical (or stationary) points where the gradient is small, or second-order critical (or stationary) points where the gradient is small and the Hessian is nearly positive semidefinite.

One of the simplest algorithms for solving (1) is gradient descent (GD): given $x_0$, iterate

$$x_{t+1} = x_t - \eta \nabla f(x_t). \tag{2}$$

It is well known that if $\nabla f$ is Lipschitz continuous, with appropriate step-size $\eta$, GD converges to first-order critical points. However, it may take exponential time to reach an approximate second-order critical point, thus, to escape *saddle points* [14]. There is an increasing amount of evidence that saddle points are a serious obstacle to the practical success of local optimization algorithms such as

GD [25, 16]. This calls for algorithms which provably escape saddle points efficiently. We focus on methods which only have access to $f$ and $\nabla f$ (but not $\nabla^2 f$) through a black-box model.

Several methods add noise to GD iterates in order to escape saddle points faster, under the assumption that $f$ has $L$-Lipschitz continuous gradient and $\rho$-Lipschitz continuous Hessian. In this setting, an *$\epsilon$-second-order critical point* is a point $x$ satisfying $\|\nabla f(x)\| \leq \epsilon$ and $\nabla^2 f(x) \succeq -\sqrt{\rho\epsilon}I$. Under the *strict saddle assumption*, with $\epsilon$ small enough, such points are near (local) minimizers [16, 17].

In 2015, Ge et al. [16] gave a variant of stochastic gradient descent (SGD) which adds isotropic noise to iterates, showing it produces an $\epsilon$-second-order critical point with high probability in $O(\text{poly}(d)/\epsilon^4)$ stochastic gradient queries. In 2017, Jin et al. [17] presented a variant of GD, perturbed gradient descent (PGD), which reduces this complexity to $O((\log d)^4/\epsilon^2)$ full gradient queries. Recently, Jin et al. [18] simplified their own analysis of PGD, and extended it to stochastic gradient descent.

Jin et al.'s PGD [18, Alg. 4] works as follows: If the gradient is large at iterate $x_t$, $\|\nabla f(x_t)\| > \epsilon$, then perform a gradient descent step: $x_{t+1} = x_t - \eta\nabla f(x_t)$. If the gradient is small at iterate $x_t$, $\|\nabla f(x_t)\| \leq \epsilon$, perturb $x_t$ by $\eta\xi$, with $\xi$ sampled uniformly from a ball of fixed radius centered at zero. Starting from this new point $x_t + \eta\xi$, perform $\mathscr{T}$ gradient descent steps, arriving at iterate $x_{t+\mathscr{T}}$. From here, repeat this procedure starting at $x_{t+\mathscr{T}}$. Crucially, Jin et al. [18] show that, if $x_t$ is not an $\epsilon$-second-order critical point, then the function decreases enough from $x_t$ to $x_{t+\mathscr{T}}$ with high probability, leading to an escape.

In this paper we generalize PGD to optimization problems on manifolds, i.e., problems of the form

$$\min_{x\in\mathcal{M}} f(x) \tag{3}$$

where $\mathcal{M}$ is an arbitrary Riemannian manifold and $f\colon \mathcal{M} \to \mathbb{R}$ is sufficiently smooth [3]. Optimization on manifolds notably occurs in machine learning (e.g., PCA [35], low-rank matrix completion [12]), computer vision (e.g., [32]) and signal processing (e.g., [2])—see [4] for more. See [29] and [26] for examples of the strict saddle property on manifolds.

Given $x \in \mathcal{M}$, the (Riemannian) gradient of $f$ at $x$, $\text{grad } f(x)$, is a vector in the tangent space at $x$, $\mathrm{T}_x\mathcal{M}$. To perform gradient descent on a manifold, we need a way to move on the manifold along the direction of the gradient at $x$. This is provided by a *retraction* $\text{Retr}_x$: a smooth map from $\mathrm{T}_x\mathcal{M}$ to $\mathcal{M}$. Riemannian gradient descent (RGD) performs steps on $\mathcal{M}$ of the form

$$x_{t+1} = \text{Retr}_{x_t}(-\eta\text{grad } f(x_t)). \tag{4}$$

For Euclidean space, $\mathcal{M} = \mathbb{R}^d$, the standard retraction is $\text{Retr}_x(s) = x + s$, in which case (4) reduces to (2). For the sphere embedded in Euclidean space, $\mathcal{M} = S^d \subset \mathbb{R}^{d+1}$, a natural retraction is given by metric projection to the sphere: $\text{Retr}_x(s) = (x + s)/\|x + s\|$.

For $x \in \mathcal{M}$, define the *pullback* $\hat{f}_x = f \circ \text{Retr}_x\colon \mathrm{T}_x\mathcal{M} \to \mathbb{R}$, conveniently defined on a linear space. If $\text{Retr}$ is nice enough (details below), the Riemannian gradient and Hessian of $f$ at $x$ equal the (classical) gradient and Hessian of $\hat{f}_x$ at the origin of $\mathrm{T}_x\mathcal{M}$. Since $\mathrm{T}_x\mathcal{M}$ is a vector space, if we perform GD on $\hat{f}_x$, we can almost directly apply Jin et al.'s analysis [18]. This motivates the two-phase structure of our *perturbed Riemannian gradient descent* (PRGD), listed as Algorithm 1.

Our PRGD is a variant of RGD (4) and a generalization of PGD. It works as follows: If the gradient is large at iterate $x_t \in \mathcal{M}$, $\|\text{grad } f(x_t)\| > \epsilon$, perform an RGD step: $x_{t+1} = \text{Retr}_{x_t}(-\eta\text{grad } f(x_t))$. We call this a "step on the manifold." If the gradient at iterate $x_t$ is small, $\|\text{grad } f(x_t)\| \leq \epsilon$, then *perturb in the tangent space* $\mathrm{T}_{x_t}\mathcal{M}$. After this perturbation, execute at most $\mathscr{T}$ gradient descent steps *on the pullback* $\hat{f}_{x_t}$, in the tangent space. We call these "tangent space steps." We denote this sequence of $\mathscr{T}$ tangent space steps by $\{s_j\}_{j\geq 0}$. This sequence of steps is performed by TANGENTSPACESTEPS: a deterministic, vector-space procedure—see Algorithm 1.

By distinguishing between gradient descent steps on the manifold and those in a tangent space, we can apply Jin et al.'s analysis almost directly [18], allowing us to prove PRGD reaches an $\epsilon$-second-order critical point on $\mathcal{M}$ in $O((\log d)^4/\epsilon^2)$ gradient queries. Regarding regularity of $f$, we require its pullbacks to satisfy Lipschitz-type conditions, as advocated in [11, 7]. The analysis is far less technical than if one runs all steps on the manifold. We expect that this two-phase approach may prove useful for the generalization of other algorithms and analyses from the Euclidean to the Riemannian realm.

Recently, Sun and Fazel [30] provided the first generalization of PGD to certain manifolds with a polylogarithmic complexity in the dimension, improving earlier results by Ge et al. [16, App. B] which had a polynomial complexity. Both of these works focus on submanifolds of a Euclidean space, with the algorithm in [30] depending on the equality constraints chosen to describe this submanifold.

At the same time as the present paper, Sun et al. [31] improved their analysis to cover any complete Riemannian manifold with bounded sectional curvature. In contrast to ours, their algorithm executes all steps on the manifold. Their analysis requires the retraction to be the Riemannian exponential map (i.e., geodesics). Our regularity assumptions are similar but different: while we assume Lipschitz-type conditions on the pullbacks in small balls around the origins of tangent spaces, Sun et al. make Lipschitz assumptions on the cost function directly, using parallel transport and Riemannian distance. As a result, curvature appears in their results. We make no explicit assumptions on $\mathcal{M}$ regarding curvature or completeness, though these may be implicit in our regularity assumptions: see Section 4.

---

**Algorithm 1** $\mathrm{PRGD}(x_0, \eta, r, \mathcal{T}, \epsilon, T, b)$

1: $t \leftarrow 0$
2: **while** $t \leq T$ **do**
3:     **if** $\|\mathrm{grad}\, f(x_t)\| > \epsilon$ **then**
4:         $x_{t+1} \leftarrow \text{TANGENTSPACESTEPS}(x_t, 0, \eta, b, 1)$         ▷ Riemannian gradient descent step
5:         $t \leftarrow t + 1$
6:     **else**
7:         $\xi \sim \mathrm{Uniform}(B_{x_t, r}(0))$                                         ▷ perturb
8:         $s_0 = \eta \xi$
9:         $x_{t+\mathcal{T}} \leftarrow \text{TANGENTSPACESTEPS}(x_t, s_0, \eta, b, \mathcal{T})$         ▷ perform $\mathcal{T}$ steps in $\mathrm{T}_{x_t}\mathcal{M}$
10:         $t \leftarrow t + \mathcal{T}$
11:     **end if**
12: **end while**
13:
14: **procedure** TANGENTSPACESTEPS$(x, s_0, \eta, b, \mathcal{T})$
15:     **for** $j = 0, 1, \ldots, \mathcal{T} - 1$ **do**
16:         $s_{j+1} \leftarrow s_j - \eta \nabla \hat{f}_x(s_j)$
17:         **if** $\|s_{j+1}\| \geq b$ **then**                       ▷ if the iterate leaves the interior of the ball $B_{x,b}(0)$
18:             $s_{\mathcal{T}} \leftarrow s_j - \alpha \eta \nabla \hat{f}_x(s_j)$, where $\alpha \in (0, 1]$ and $\left\| s_j - \alpha \eta \nabla \hat{f}_x(s_j) \right\| = b$.
19:             **break**
20:         **end if**
21:     **end for**
22:     **return** $\mathrm{Retr}_x(s_{\mathcal{T}})$
23: **end procedure**

---

## 1.1 Main result

Here we state our result informally. Formal results are stated in subsequent sections.

**Theorem 1.1** (Informal). *Let $\mathcal{M}$ be a Riemannian manifold of dimension $d$ equipped with a retraction* $\mathrm{Retr}$. *Assume $f\colon \mathcal{M} \to \mathbb{R}$ is twice continuously differentiable, and furthermore:*

    *A1. $f$ is lower bounded.*

    *A2. The gradients of the pullbacks $f \circ \mathrm{Retr}_x$ uniformly satisfy a Lipschitz-type condition.*

    *A3. The Hessians of the pullbacks $f \circ \mathrm{Retr}_x$ uniformly satisfy a Lipschitz-type condition.*

    *A4. The retraction $\mathrm{Retr}$ uniformly satisfies a second-order condition.*

*Then, setting $T = O((\log d)^4 / \epsilon^2)$, PRGD visits several points with gradient smaller than $\epsilon$ and, with high probability, at least two-thirds of those points are $\epsilon$-second-order critical (Definition 3.1).*

PRGD uses $O((\log d)^4 / \epsilon^2)$ gradient queries, and crucially no Hessian queries. The algorithm requires knowledge of the Lipschitz constants defined below, which makes this a mostly theoretical algorithm—but see Appendix D for explicit constants in the case of PCA.

## 1.2 Other related work

Algorithms which efficiently escape saddle points can be classified into two families: first-order and second-order methods. First-order methods only use function value and gradient information. SGD and PGD are first-order methods. Second-order methods also access Hessian information. Newton's method, trust regions [24, 11] and adaptive cubic regularization [23, 7, 34] are second-order methods.

As noted above, Ge et al. [16] and Jin et al. [17] escape saddle points (in Euclidean space) by exploiting noise in iterations. There has also been similar work for normalized gradient descent [20]. Expanding on [17], Jin et al. [19] give an accelerated PGD algorithm (PAGD) which reaches an $\epsilon$-second-order critical point of a non-convex function $f$ with high probability in $O((\log d)^6/\epsilon^{7/4})$ iterations. In [18], Jin et al. show that a stochastic version of PGD reaches an $\epsilon$-second-order critical point in $O(d/\epsilon^4)$ stochastic gradient queries; only $O(\text{poly}(\log d)/\epsilon^4)$ queries are needed if the stochastic gradients are well behaved. For an analysis of PGD under convex constraints, see [22].

There is another line of research, inspired by Langevin dynamics, in which judiciously scaled Gaussian noise is added at every iteration. We note that although this differs from the first incarnation of PGD in [17], this resembles a simplified version of PGD in [18]. Sang and Liu [27] develop an algorithm (adaptive stochastic gradient Langevin dynamics, ASGLD), which provably reaches an $\epsilon$-second-order critical point in $O(\log d/\epsilon^4)$ with high probability. With full gradients, AGSLD reaches an $\epsilon$-second-order critical point in $O(\log d/\epsilon^2)$ queries with high probability.

One might hope that the noise inherent in vanilla SGD would help it escape saddle points without noise injection. Daneshmand et al. [13] propose the correlated negative curvature assumption (CNC), under which they prove that SGD reaches an $\epsilon$-second-order critical point in $O(\epsilon^{-5})$ queries with high probability. They also show that, under the CNC assumption, a variant of GD (in which iterates are perturbed only by SGD steps) efficiently escapes saddle points. Importantly, these guarantees are completely dimension-free.

A first-order method can include approximations of the Hessian (e.g., with a difference of gradients). For example, Allen-Zhu's Natasha 2 algorithm [8] uses first-order information (function value and stochastic gradients) to search for directions of negative curvature of the Hessian. Natasha 2 reaches an $\epsilon$-second-order critical point in $O(\epsilon^{-13/4})$ iterations.

Many classical optimization algorithms have been generalized to optimization on manifolds, including gradient descent, Newton's method, trust regions and adaptive cubic regularization [15, 3, 1, 6, 11, 7, 9, 34]. Bonnabel [10] extends stochastic gradient descent to Riemannian manifolds and proves that Riemannian SGD converges to critical points of the cost function. Zhang et al. [33] and Sato et al. [28] both use variance reduction to speed up SGD on Riemannian manifolds.

## 2 Preliminaries: Optimization on manifolds

We review the key definitions and tools for optimization on manifolds. For more information, see [3]. Let $\mathcal{M}$ be a $d$-dimensional Riemannian manifold: a real, smooth $d$-manifold equipped with a Riemannian metric. We associate with each $x \in \mathcal{M}$ a $d$-dimensional real vector space $\mathrm{T}_x\mathcal{M}$, called the tangent space at $x$. For embedded submanifolds of $\mathbb{R}^n$, we often visualize the tangent space as being tangent to the manifold at $x$. The Riemannian metric defines an inner product $\langle \cdot, \cdot \rangle_x$ on the tangent space $\mathrm{T}_x\mathcal{M}$, with associated norm $\|\cdot\|_x$. We denote these by $\langle \cdot, \cdot \rangle$ and $\|\cdot\|$ when $x$ is clear from context. A vector in the tangent space is a tangent vector. The set of pairs $(x, s_x)$ for $x \in \mathcal{M}, s_x \in \mathrm{T}_x\mathcal{M}$ is called the tangent bundle $\mathrm{T}\mathcal{M}$. Define $B_{x,r}(s) = \{\dot{s} \in \mathrm{T}_x\mathcal{M} : \|\dot{s} - s\|_x \leq r\}$: the closed ball of radius $r$ centered at $s \in \mathrm{T}_x\mathcal{M}$. We occasionally denote $B_{x,r}(s)$ by $B_r(s)$ when $x$ is clear from context. Let $\text{Uniform}(B_{x,r}(s))$ denote the uniform distribution over the ball $B_{x,r}(s)$.

The Riemannian gradient $\text{grad} f(x)$ of a differentiable function $f$ at $x \in \mathcal{M}$ is the unique vector in $\mathrm{T}_x\mathcal{M}$ satisfying $\mathrm{D}f(x)[s] = \langle \text{grad} f(x), s \rangle_x \; \forall s \in \mathrm{T}_x\mathcal{M}$, where $\mathrm{D}f(x)[s]$ is the directional derivative of $f$ at $x$ along $s$. The Riemannian metric gives rise to a well-defined notion of derivative of vector fields called the Riemannian (or Levi–Civita) connection $\nabla$. The Hessian of $f$ is the derivative of the gradient vector field: $\text{Hess} f(x)[u] = \nabla_u \text{grad} f(x)$. The Hessian describes how the gradient changes. $\text{Hess} f(x)$ is a symmetric linear operator on $\mathrm{T}_x\mathcal{M}$. If the manifold is a Euclidean space, $\mathcal{M} = \mathbb{R}^d$, with the standard metric $\langle x, y \rangle = x^T y$, the Riemannian gradient $\text{grad} f$ and Hessian $\text{Hess} f$ coincide with the standard gradient $\nabla f$ and Hessian $\nabla^2 f$ (mind the overloaded notation $\nabla$).

As discussed in Section 1, the retraction is a mapping which allows us to move along the manifold from a point $x$ in the direction of a tangent vector $s \in \mathrm{T}_x\mathcal{M}$. Formally:

**Definition 2.1** (Retraction, from [3]). *A retraction on a manifold $\mathcal{M}$ is a smooth mapping $\mathrm{Retr}$ from the tangent bundle $\mathrm{T}\mathcal{M}$ to $\mathcal{M}$ satisfying properties 1 and 2 below. Let $\mathrm{Retr}_x \colon \mathrm{T}_x\mathcal{M} \to \mathcal{M}$ denote the restriction of $\mathrm{Retr}$ to $\mathrm{T}_x\mathcal{M}$.*

  1. *$\mathrm{Retr}_x(0_x) = x$, where $0_x$ is the zero vector in $\mathrm{T}_x\mathcal{M}$.*

  2. *The differential of $\mathrm{Retr}_x$ at $0_x$, $\mathrm{DRetr}_x(0_x)$, is the identity map.*

(Our algorithm and theory only require $\mathrm{Retr}$ to be defined in balls of a fixed radius around the origins of tangent spaces.) Recall these special retractions, which are good to keep in mind for intuition: on $\mathcal{M} = \mathbb{R}^d$, we typically use $\mathrm{Retr}_x(s) = x + s$, and on the unit sphere we typically use $\mathrm{Retr}_x(s) = (x + s)/\left\| x + s \right\|$.

For $x$ in $\mathcal{M}$, define the pullback of $f$ from the manifold to the tangent space by

$$\hat{f}_x = f \circ \mathrm{Retr}_x \colon \mathrm{T}_x\mathcal{M} \to \mathbb{R}.$$

This is a real function on a vector space. Furthermore, for $x \in \mathcal{M}$ and $s \in \mathrm{T}_x\mathcal{M}$, let

$$T_{x,s} = \mathrm{DRetr}_x(s) \colon \mathrm{T}_x\mathcal{M} \to \mathrm{T}_{\mathrm{Retr}_x(s)}\mathcal{M}$$

denote the differential of $\mathrm{Retr}_x$ at $s$ (a linear operator). The gradient and Hessian of the pullback admit the following nice expressions in terms of those of $f$, and the retraction.

**Lemma 2.2** (Lemma 5.2 of [7]). *For $f \colon \mathcal{M} \to \mathbb{R}$ twice continuously differentiable, $x \in \mathcal{M}$ and $s \in \mathrm{T}_x\mathcal{M}$, with $T_{x,s}^*$ denoting the adjoint of $T_{x,s}$,*

$$\nabla \hat{f}_x(s) = T_{x,s}^* \mathrm{grad}\, f(\mathrm{Retr}_x(s)), \qquad \nabla^2 \hat{f}_x(s) = T_{x,s}^* \mathrm{Hess}\, f(\mathrm{Retr}_x(s)) T_{x,s} + W_s, \quad (5)$$

*where $W_s$ is a symmetric linear operator on $\mathrm{T}_x\mathcal{M}$ defined through polarization by*

$$\langle W_s[\dot{s}], \dot{s} \rangle = \langle \mathrm{grad}\, f(\mathrm{Retr}_x(s)), \gamma''(0) \rangle, \tag{6}$$

*with $\gamma''(0) \in \mathrm{T}_{\mathrm{Retr}_x(s)}\mathcal{M}$ the intrinsic acceleration on $\mathcal{M}$ of $\gamma(t) = \mathrm{Retr}_x(s + t\dot{s})$ at $t = 0$.*

The velocity of a curve $\gamma \colon \mathbb{R} \to \mathcal{M}$ is $\frac{d\gamma}{dt} = \gamma'(t)$. The intrinsic acceleration $\gamma''$ of $\gamma$ is the covariant derivative (induced by the Levi–Civita connection) of the velocity of $\gamma$: $\gamma'' = \frac{\mathrm{D}}{dt}\gamma'$. When $\mathcal{M}$ is a Riemannian submanifold of $\mathbb{R}^n$, $\gamma''(t)$ does not necessarily coincide with $\frac{d^2\gamma}{dt^2}$: in this case, $\gamma''(t)$ is the orthogonal projection of $\frac{d^2\gamma}{dt^2}$ onto $\mathrm{T}_{\gamma(t)}\mathcal{M}$.

## 3 PRGD efficiently escapes saddle points

We now precisely state the assumptions, the main result, and some important parts of the proof of the main result, including the main obstacles faced in generalizing PGD to manifolds. A full proof of all results is provided in the appendix.

### 3.1 Assumptions

The first assumption, namely, that $f$ is lower bounded, ensures that there are points on the manifold where the gradient is arbitrarily small.

**Assumption 1.** *$f$ is lower bounded: $f(x) \geq f^*$ for all $x \in \mathcal{M}$.*

Generalizing from the Euclidean case, we assume Lipschitz-type conditions on the gradients and Hessians of the pullbacks $\hat{f}_x = f \circ \mathrm{Retr}_x$. For the special case of $\mathcal{M} = \mathbb{R}^d$ and $\mathrm{Retr}_x(s) = x + s$, these assumptions hold if the gradient $\nabla f(\cdot)$ and Hessian $\nabla^2 f(\cdot)$ are each Lipschitz continuous, as in [18, A1] (with the same constants). The Lipschitz-type assumptions below are similar to assumption A2 of [7]. Notice that these assumptions involve both the cost function and the retraction: this dependency is further discussed in [11, 7] for a similar setting.

**Assumption 2.** *There exist $b_1 > 0$ and $L > 0$ such that $\forall x \in \mathcal{M}$ and $\forall s \in \mathrm{T}_x\mathcal{M}$ with $\|s\| \leq b_1$,*

$$\left\| \nabla \hat{f}_x(s) - \nabla \hat{f}_x(0) \right\| \leq L \|s\| .$$

**Assumption 3.** *There exist $b_2 > 0$ and $\rho > 0$ such that $\forall x \in \mathcal{M}$ and $\forall s \in \mathrm{T}_x\mathcal{M}$ with $\|s\| \leq b_2$,*

$$\left\| \nabla^2 \hat{f}_x(s) - \nabla^2 \hat{f}_x(0) \right\| \leq \rho \|s\| ,$$

*where on the left-hand side we use the operator norm.*

More precisely, we only need these assumptions to hold at the iterates $x_0, x_1, \dots$ Let $b = \min\{b_1, b_2\}$ (to reduce the number of parameters in Algorithm 1). The next assumption requires the chosen retraction to be well behaved, in the sense that the (intrinsic) acceleration of curves $\gamma_{x,s}$ on the manifold, defined below, must remain bounded—compare with Lemma 2.2.

**Assumption 4.** *There exists $\beta \geq 0$ such that, for all $x \in \mathcal{M}$ and $s \in \mathrm{T}_x\mathcal{M}$ satisfying $\|s\| = 1$, the curve $\gamma_{x,s}(t) = \mathrm{Retr}_x(ts)$ has initial acceleration bounded by $\beta$: $\left\| \gamma''_{x,s}(0) \right\| \leq \beta$.*

If Assumption 4 holds with $\beta = 0$, $\mathrm{Retr}$ is said to be *second order* [3, p107]. Second-order retractions include the so-called exponential map and the standard retractions on $\mathbb{R}^d$ and the unit sphere mentioned earlier—see [5] for a large class of such retractions on relevant manifolds.

**Definition 3.1.** *A point $x \in \mathcal{M}$ is an $\epsilon$-second-order critical point of the twice-differentiable function $f \colon \mathcal{M} \to \mathbb{R}$ satisfying Assumption 3 if*

$$\|\mathrm{grad}\, f(x)\| \leq \epsilon, \qquad \text{and} \qquad \lambda_{\min}(\mathrm{Hess}\, f(x)) \geq -\sqrt{\rho\epsilon}, \tag{7}$$

*where $\lambda_{\min}(H)$ denotes the smallest eigenvalue of the symmetric operator $H$.*

For compact manifolds, all of these assumptions hold (all proofs are in the appendix):

**Lemma 3.2.** *Let $\mathcal{M}$ be a compact Riemannian manifold equipped with a retraction $\mathrm{Retr}$. Assume $f \colon \mathcal{M} \to \mathbb{R}$ is three times continuously differentiable. Pick an arbitrary $b > 0$. Then, there exist $f^*, L > 0, \rho > 0$ and $\beta \geq 0$ such that Assumptions 1, 2, 3 and 4 are satisfied.*

## 3.2 Main results

Recall that PRGD (Algorithm 1) works as follows. If $\|\mathrm{grad}\, f(x_t)\| > \epsilon$, perform a Riemannian gradient descent step, $x_{t+1} = \mathrm{Retr}_{x_t}(-\eta\mathrm{grad}\, f(x_t))$. If $\|\mathrm{grad}\, f(x_t)\| \leq \epsilon$, then perturb, i.e., sample $\xi \sim \mathrm{Uniform}(B_{x_t, r}(0))$ and let $s_0 = \eta\xi$. After this perturbation, remain in the tangent space $\mathrm{T}_{x_t}\mathcal{M}$ and do (at most) $\mathscr{T}$ gradient descent steps on the pullback $\hat{f}_{x_t}$, starting from $s_0$. We denote this sequence of $\mathscr{T}$ tangent space steps by $\{s_j\}_{j \geq 0}$. This sequence of gradient descent steps is performed by TANGENTSPACESTEPS: a deterministic procedure in the (linear) tangent space.

One difficulty with this approach is that, under our assumptions, for some $x = x_t$, $\nabla \hat{f}_x$ may not be Lipschitz continuous in all of $\mathrm{T}_x\mathcal{M}$. However, it is easy to show that $\nabla \hat{f}_x$ is Lipschitz continuous in the ball of radius $b$ by compactness, uniformly in $x$. This is why we limit our algorithm to these balls. If the sequence of iterates $\{s_j\}_{j \geq 0}$ escapes the ball $B_{x,b}(0) \subset \mathrm{T}_x\mathcal{M}$ for some $s_j$, TANGENTSPACESTEPS returns the point between $s_{j-1}$ and $s_j$ on the boundary of that ball.

Following [18], we use a set of carefully balanced parameters. Parameters $\epsilon$ and $\delta$ are user defined. The claim in Theorem 3.4 below holds with probability at least $1 - \delta$. Assumption 1 provides parameter $f^*$. Assumptions 2 and 3 provide parameters $L, \rho$ and $b = \min\{b_1, b_2\}$. As announced, the latter two assumptions further ensure Lipschitz continuity of the gradients of the pullbacks in balls of the tangent spaces, uniformly: this defines the parameter $\ell$, as prescribed below.

**Lemma 3.3.** *Under Assumptions 2 and 3, there exists $\ell \in [L, L + \rho b]$ such that, for all $x \in \mathcal{M}$, the gradient of the pullback, $\nabla \hat{f}_x$, is $\ell$-Lipschitz continuous in the ball $B_{x,b}(0)$.*

Then, choose $\chi > 1/4$ (preferably small) such that

$$\chi \geq 4 \log_2 \left( 2^{31} \frac{\ell^2 \sqrt{d}(f(x_0) - f^*)}{\delta \sqrt{\rho} \epsilon^{5/2}} \right), \tag{8}$$

and set algorithm parameters

$$\eta = \frac{1}{\ell}, \qquad\qquad r = \frac{\epsilon}{400\chi^3}, \qquad\qquad \mathscr{T} = \frac{\ell\chi}{\sqrt{\rho\epsilon}}, \qquad (9)$$

where $\chi$ is such that $\mathscr{T}$ is an integer. We also use this notation in the proofs:

$$\mathscr{F} = \frac{1}{50\chi^3}\sqrt{\frac{\epsilon^3}{\rho}}, \qquad\qquad\qquad \mathscr{L} = \frac{1}{4\chi}\sqrt{\frac{\epsilon}{\rho}}. \qquad (10)$$

**Theorem 3.4.** *Assume $f$ satisfies Assumptions 1, 2 and 3. For any $x_0 \in \mathcal{M}$, with $0 < \epsilon \leq b^2\rho$, $L \geq \sqrt{\rho\epsilon}$, $\epsilon^{3/2} \leq 3\sqrt{\rho}\,(f(x_0) - f^*)$ and $\delta \in (0,1)$, choose $\eta, r, \mathscr{T}$ as in (9). Then, setting*

$$T = 8\max\left\{\frac{\mathscr{T}}{3}, \frac{(f(x_0) - f^*)\mathscr{T}}{\mathscr{F}}, \frac{f(x_0) - f^*}{\eta\epsilon^2}\right\} = O\!\left(\frac{\ell(f(x_0) - f^*)}{\epsilon^2}(\log d)^4\right), \qquad (11)$$

*$PRGD(x_0, \eta, r, \mathscr{T}, \epsilon, T, b)$ visits at least two iterates $x_t \in \mathcal{M}$ satisfying $\|\mathrm{grad}\, f(x_t)\| \leq \epsilon$. With probability at least $1 - \delta$, at least two-thirds of those iterates satisfy*

$$\|\mathrm{grad}\, f(x_t)\| \leq \epsilon \qquad\qquad and \qquad\qquad \lambda_{\min}(\nabla^2 \hat{f}_{x_t}(0)) \geq -\sqrt{\rho\epsilon}.$$

*The algorithm uses at most $T + \mathscr{T} \leq 2T$ gradient queries (and no function or Hessian queries).*

By Assumption 4 and Lemma 2.2, $\nabla^2 \hat{f}_{x_t}(0)$ is close to $\mathrm{Hess}\, f(x_t)$, which allows us to conclude:

**Corollary 3.5.** *Assume $f$ satisfies Assumptions 1, 2, 3 and 4. For an arbitrary $x_0 \in \mathcal{M}$, with $0 < \epsilon \leq \min\{\rho/\beta^2, b^2\rho\}$, $L \geq \sqrt{\rho\epsilon}$, $\epsilon^{3/2} \leq 3\sqrt{\rho}\,(f(x_0) - f^*)$ and $\delta \in (0,1)$, choose $\eta, r, \mathscr{T}$ as in (9). Then, setting $T$ as in (11), $PRGD(x_0, \eta, r, \mathscr{T}, \epsilon, T, b)$ visits at least two iterates $x_t \in \mathcal{M}$ satisfying $\|\mathrm{grad}\, f(x_t)\| \leq \epsilon$. With probability at least $1 - \delta$, at least two-thirds of those iterates are $(4\epsilon)$-second-order points. If $\beta = 0$ (that is, the retraction is second order), then the same claim holds for $\epsilon$-second-order points instead of $4\epsilon$. The algorithm uses at most $T + \mathscr{T} \leq 2T$ gradient queries.*

Assume $\mathcal{M} = \mathbb{R}^d$ with standard inner product and standard retraction $\mathrm{Retr}_x(s) = x + s$. As in [18], assume $f\colon \mathbb{R}^d \to \mathbb{R}$ is lower bounded, $\nabla f$ is $L$-Lipschitz in $\mathbb{R}^d$, and $\nabla^2 f$ is $\rho$-Lipschitz in $\mathbb{R}^d$. Then, Assumptions 1, 2 and 3 hold with $b = +\infty$. Furthermore, Assumption 4 holds with $\beta = 0$ so that $\nabla^2 \hat{f}_x(0) = \mathrm{Hess}\, f(x) = \nabla^2 f(x)$ (Lemma 2.2). For all $x \in \mathcal{M}$, $\nabla \hat{f}_x(s)$ has Lipschitz constant $\ell = L$ since $\hat{f}_x(s) = f(x + s)$. Therefore, using $b = +\infty$, $\ell = L$ and choosing $\eta, r, \mathscr{T}$ as in (9), PRGD reduces to PGD, and Theorem 3.4 recovers the result of Jin et al. [18]: this confirms that the present result is a bona fide generalization.

For the important special case of compact manifolds, Lemmas 3.2 and 3.3 yield:

**Corollary 3.6.** *Assume $\mathcal{M}$ is a compact Riemannian manifold equipped with a retraction $\mathrm{Retr}$, and $f\colon \mathcal{M} \to \mathbb{R}$ is three times continuously differentiable. Pick an arbitrary $b > 0$. Then, Assumptions 1, 2, 3, 4 hold for some $L > 0$, $\rho > 0$, $\beta \geq 0$, so that Corollary 3.5 applies with some $\ell \in [L, L + \rho b]$.*

**Remark 3.7.** *PRGD, like PGD (Algorithm 4 in [18]), does not specify which iterate is an $\epsilon$-second-order critical point. However, it is straightforward to include a termination condition in PRGD which halts the algorithm and returns a suspected $\epsilon$-second-order critical point. Indeed, Jin et al. include such a termination condition in their original PGD algorithm [17], which here would go as follows: After performing a perturbation and $\mathscr{T}$ (tangent space) steps in $\mathrm{T}_{x_t}\mathcal{M}$, return $x_t$ if $\hat{f}_{x_t}(s_{\mathscr{T}}) - \hat{f}_{x_t}(0) > -f_{\mathrm{thres}}$, i.e., the function value does not decrease enough. The termination condition requires a threshold $f_{\mathrm{thres}}$ which is balanced like the other parameters of PRGD in (9).*

### 3.3 Main proof ideas

Theorem 3.4 follows from the following two lemmas which we prove in the appendix. These lemmas state that, in each round of the while-loop in PRGD, if $x_t$ is not at an $\epsilon$-second-order critical point, PRGD makes progress, that is, decreases the cost function value (the first lemma is deterministic, the second one is probabilistic). Yet, the value of $f$ on the iterates can only decrease so much because $f$ is bounded below by $f^*$. Therefore, the probability that PRGD does not visit an $\epsilon$-second-order critical point is low.

**Lemma 3.8.** *Under Assumptions 2 and 3, set $\eta = 1/\ell$ for some $\ell \geq L$. If $x \in \mathcal{M}$ satisfies $\|\mathrm{grad}\, f(x)\| > \epsilon$ with $\epsilon \leq b^2 \rho$ and $L \geq \sqrt{\rho\epsilon}$, then,*

$$f(\textsc{TangentSpaceSteps}(x, 0, \eta, b, 1)) - f(x) \leq -\eta\epsilon^2/2.$$

**Lemma 3.9.** *Under Assumptions 2 and 3, let $x \in \mathcal{M}$ satisfy both $\|\mathrm{grad}\, f(x)\| \leq \epsilon$ and $\lambda_{\min}(\nabla^2 \hat{f}_x(0)) \leq -\sqrt{\rho\epsilon}$ with $\epsilon \leq b^2 \rho$ and $L \geq \sqrt{\rho\epsilon}$. Set $\eta, r, \mathcal{T}, \mathcal{F}$ as in (9) and (10). Let $s_0 = \eta\xi$ with $\xi \sim Uniform(B_{x,r}(0))$. Then,*

$$\mathbb{P}\big[f(\textsc{TangentSpaceSteps}(x, s_0, \eta, b, \mathcal{T})) - f(x) \leq -\mathcal{F}/2\big] \geq 1 - \frac{\ell\sqrt{d}}{\sqrt{\rho\epsilon}} 2^{10-\chi/2}.$$

Lemma 3.8 states that we are guaranteed to make progress if the gradient is large. This follows from the sufficient decrease of RGD steps. Lemma 3.9 states that, with perturbation, GD on the pullback escapes a saddle point with high probability. Lemma 3.9 is analogous to Lemma 11 in [18].

Let $\mathcal{X}_{\mathrm{stuck}}$ be the set of tangent vectors $s_0$ in $B_{x,\eta r}(0)$ for which GD on the pullback starting from $s_0$ does not escape the saddle point, i.e., the function value does not decrease enough after $\mathcal{T}$ iterations. Following Jin et al.'s analysis [18], we bound the width of this "stuck region" (in the direction of the eigenvector $e_1$ associated with the minimum eigenvalue of the Hessian of the pullback, $\nabla^2 \hat{f}_x(0)$). Like Jin et al., we do this with a coupling argument, showing that given two GD sequences with starting points sufficiently far apart, one of these sequences must escape. This is formalized in Lemma C.4 of the appendix. A crucial observation to prove Lemma C.4 is that, if the function value of GD iterates does not decrease much, then these iterates must be localized; this is formalized in Lemma C.3 of the appendix, which Jin et al. call "improve or localize."

We stress that the stuck region concept, coupling argument, improve or local paradigm, and details of the analysis are due to Jin et al. [18]: our main contribution is to show a clean way to generalize the algorithm to manifolds in such a way that the analysis extends with little friction. We believe that the general idea of separating iterations between the manifold and the tangent spaces to achieve different objectives may prove useful to generalize other algorithms as well.

## 4 About the role of curvature of the manifold

As pointed out in the introduction, concurrently with our work, Sun et al. [31] have proposed another generalization of PGD to manifolds. Their algorithm executes all steps on the manifold directly (as opposed to our own, which makes certain steps in the tangent spaces), and moves around the manifold using the exponential map. To carry out their analysis, Sun et al. assume $f$ is regular in the following way. The Riemannian gradient is Lipschitz continuous in a Riemannian sense, namely,

$$\forall x, y \in \mathcal{M}, \qquad \|\mathrm{grad}\, f(y) - \Gamma_x^y \mathrm{grad}\, f(x)\| \leq L\mathrm{dist}(x, y),$$

where $\Gamma_x^y \colon \mathrm{T}_x\mathcal{M} \to \mathrm{T}_y\mathcal{M}$ denotes parallel transport from $x$ to $y$ along any minimizing geodesic, and $\mathrm{dist}$ is the Riemannian distance. These notions are well defined if $\mathcal{M}$ is a connected, complete manifold. Similarly, they assume the Riemannian Hessian of $f$ is Lipschitz continuous in a Riemannian sense:

$$\forall x, y \in \mathcal{M}, \qquad \|\mathrm{Hess}\, f(y) - \Gamma_x^y \circ \mathrm{Hess}\, f(x) \circ \Gamma_y^x\| \leq \rho\mathrm{dist}(x, y),$$

in the operator norm. Using (and improving) sophisticated inequalities from Riemannian geometry, they map the perturbed sequences back to tangent spaces for analysis, where they run an adapted version of Jin et al.'s argument. In so doing, it appears to be crucial to use the exponential map, owing to its favorable interplay with parallel transport along geodesics and Riemannian distance, providing a good fit with the regularity conditions above.

As they map sequences back from the manifold to a common tangent space through the inverse of the exponential map, the Riemannian curvature of the manifold comes into play. Consequently, they assume $\mathcal{M}$ has bounded sectional curvature (both from below and from above), and these bounds on curvature come up in their final complexity result: constants degrade if the manifold is more curved.

Since Riemannian curvature does not occur in our own complexity result for PRGD, it is legitimate to ask: is curvature supposed to occur? If so, it must be hidden in our analysis, for example in the

regularity assumptions we make, which are expressed in terms of pullbacks rather than with parallel transports. And indeed, in several attempts to deduce our own assumptions from those of Sun et al., invariably, we had to degrade $L$ and $\rho$ as a function of curvature—minding that these are only bounds. On the other hand, under the assumptions of Sun et al., one can deduce that the regularity assumptions required in [11, 7] for the analysis of Riemannian gradient descent, trust regions and adaptive regularization by cubics hold with the exponential map, leading to curvature-free complexity bounds for all three algorithms. Thus, it is not clear that curvature should occur.

We believe this poses an interesting question regarding the complexity of optimization on manifolds: to what extent should it be influenced by curvature of the manifold? We intend to study this.

## 5   Perspectives

To perform PGD (Algorithm 4 of [18]), one must know the step size $\eta$, perturbation radius $r$ and the number of steps $\mathscr{T}$ to perform after perturbation. These parameters are carefully balanced, and their values depend on the smoothness parameters $L$ and $\rho$. In most situations, we do not know $L$ or $\rho$ (though see Appendix D for PCA). An algorithm which does not require knowledge of $L$ or $\rho$ but still has the same guarantees as PGD would be useful. However, that certain regularity parameters must be known seems inevitable, in particular for the Hessian's $\rho$. Indeed, the main theorems make statements about the spectrum of the Hessian, yet the algorithm is not allowed to query the Hessian.

GD equipped with a backtracking line-search method achieves an $\epsilon$-first-order critical point in $O(\epsilon^{-2})$ gradient queries without knowledge of the Lipschitz constant $L$. At each iterate $x_t$ of GD, backtracking line-search essentially uses function and gradient queries to estimate the gradient Lipschitz parameter near $x_t$. Perhaps PGD can perform some kind of line-search to locally estimate $L$ and $\rho$. We note that if $\rho$ is known and we use line-search-type methods to estimate $L$, there are still difficulties applying Jin et al.'s coupling argument.

Jin et al. [18] develop a stochastic version of PGD known as PSGD. Instead of perturbing when the gradient is small and performing $\mathscr{T}$ GD steps, PSGD simply performs a stochastic gradient step and perturbation at each step. Distinguishing between manifold steps and tangent space steps, we suspect it is possible to develop a Riemannian version of perturbed stochastic gradient descent which achieves an $\epsilon$-second-order critical point in $O(d/\epsilon^4)$ stochastic gradient queries, like PSGD. This Riemannian version performs a certain number of steps in the tangent space, like PRGD.

More broadly, we anticipate that it should be possible to extend several classical optimization methods from the Euclidean case to the Riemannian case through this approach of running many steps in a given tangent space before retracting. This ought to be particularly beneficial for algorithms whose computations or analysis rely intimately on linear structures, such as for coordinate descent algorithms, certain parallelized schemes, and possibly also accelerated schemes. In preparing the final version of this paper, we found that this idea is also the subject of another paper at NeurIPS 2019, where it is called dynamic trivialization [21].

### Acknowledgments

We thank Yue Sun, Nicolas Flammarion and Maryam Fazel, authors of [31], for numerous relevant discussions. NB is partially supported by NSF grant DMS-1719558.

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
