[Supplementary Material]

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

# Appendices

## A   Proof that assumptions hold for compact manifolds

*Proof of Lemma 3.2.*  Since $\mathcal{M}$ is compact and $f$ is continuous, $f$ is lower bounded by some $f^*$.

Recall $\hat{f}_x(s) = f \circ \mathrm{Retr}_x(s)$. Define $\phi, \psi \colon \mathrm{T}\mathcal{M} \to \mathbb{R}$ using operator norms by

$$\phi(x,s) = \left\| \nabla^2 \hat{f}_x(s) \right\| = \left\| \nabla_s^2 (f \circ \mathrm{Retr}(x,s)) \right\|,$$

$$\psi(x,s) = \left\| \nabla^3 \hat{f}_x(s) \right\| = \left\| \nabla_s^3 (f \circ \mathrm{Retr}(x,s)) \right\|.$$

Since $f$ is three times continuously differentiable and $\mathrm{Retr}$ is smooth, $\phi$ and $\psi$ are each continuous on the tangent bundle $\mathrm{T}\mathcal{M}$. The set

$$S_b = \{(x,s) : x \in \mathcal{M}, s \in \mathrm{T}_x\mathcal{M} \text{ with } \|s\| \leq b\}$$

is a compact subset of the tangent bundle $\mathrm{T}\mathcal{M}$ since $\mathcal{M}$ is compact. Thus, we may define

$$L = \max_{(x,s) \in S_b} \phi(x,s), \qquad \text{and} \qquad \rho = \max_{(x,s) \in S_b} \psi(x,s),$$

so that $\left\| \nabla^2 \hat{f}_x(s) \right\| \leq L$ and $\left\| \nabla^3 \hat{f}_x(s) \right\| \leq \rho$ for all $x \in \mathcal{M}$ and $s \in B_{x,b}(0)$. From here, it is clear that Assumptions 2 and 3 are satisfied, for we can just integrate as in eq. (13) below.

Using the notation from Assumption 4, the map $\upsilon \colon \mathrm{T}\mathcal{M} \to \mathbb{R}$ given by $\upsilon(x,s) = \left\| \gamma''_{x,s}(0) \right\|$ is continuous since $\mathrm{Retr}$ is smooth. The set

$$V_b = \{(x,s) : x \in \mathcal{M}, s \in \mathrm{T}_x\mathcal{M} \text{ with } \|s\| = 1\}$$

is also compact in $\mathrm{T}\mathcal{M}$. Hence, $\beta = \max_{(x,s) \in V_b} \upsilon(x,s)$ is a valid choice. $\qquad \square$

## B   Proofs for the main results

The proof follows that of Jin et al. [18] closely, reusing many of their key lemmas: we repeat some here for convenience, while highlighting the specificities of the manifold case. We consider it a contribution of this paper that, as a result of our distinction between manifold and tangent space steps, there is limited extra friction, despite the significantly extended generality. In this section and the next, all parameters are chosen as in (9) and (10).

We assume $\epsilon \leq b^2\rho$. We also assume $L \geq \sqrt{\rho\epsilon}$ because otherwise we can reach a point satisfying $\|\mathrm{grad}\, f(x)\| \leq \epsilon$ and $\lambda_{\min}(\nabla^2 \hat{f}_x(0)) \geq -\sqrt{\rho\epsilon}$ simply using RGD. Indeed, RGD always finds a point $x \in \mathcal{M}$ satisfying $\|\mathrm{grad}\, f(x)\| \leq \epsilon$, and Assumption 2 implies $\|\nabla^2 \hat{f}_x(0)\| \leq L$ so that $\lambda_{\min}(\nabla^2 \hat{f}_x(0)) \geq -L$. Thus, if $L < \sqrt{\rho\epsilon}$, every point $x \in \mathcal{M}$ satisfies $\lambda_{\min}(\nabla^2 \hat{f}_x(0)) \geq -\sqrt{\rho\epsilon}$.

We want to prove Theorem 3.4. This theorem follows from the following two lemmas (repeated from Lemmas 3.8 and 3.9 for convenience), which we prove in Appendix C below. Lemma B.1 is deterministic: it is a statement about the cost decrease produced by a single Riemannian gradient step, with bounded step size. Lemma B.2 is probabilistic, and is analogous to Lemma 11 in [18].

**Lemma B.1.** *Under Assumptions 2 and 3, set $\eta = 1/\ell$ for some $\ell \geq L$. If $x \in \mathcal{M}$ satisfies $\|\mathrm{grad}\, f(x)\| > \epsilon$ with $\epsilon \leq b^2\rho$ and $L \geq \sqrt{\rho\epsilon}$, then,*

$$f(\textsc{TangentSpaceSteps}(x, 0, \eta, b, 1)) - f(x) \leq -\eta\epsilon^2/2.$$

**Lemma B.2.** *Under Assumptions 2 and 3, let $x \in \mathcal{M}$ satisfy both $\|\mathrm{grad}\, f(x)\| \leq \epsilon$ and $\lambda_{\min}(\nabla^2 \hat{f}_x(0)) \leq -\sqrt{\rho\epsilon}$ with $\epsilon \leq b^2\rho$ and $L \geq \sqrt{\rho\epsilon}$. Set $\eta, r, \mathscr{T}, \mathscr{F}$ as in (9) and (10). Let $s_0 = \eta\xi$ with $\xi \sim Uniform(B_{x,r}(0))$. Then,*

$$\mathbb{P}\big[ f(\textsc{TangentSpaceSteps}(x, s_0, \eta, b, \mathscr{T})) - f(x) \leq -\mathscr{F}/2 \big] \geq 1 - \frac{\ell\sqrt{d}}{\sqrt{\rho\epsilon}} 2^{10-\chi/2}.$$

*Proof of Theorem 3.4.* This proof is similar to Jin et al.'s proof of Theorem 9 in [18].

Recall that we set

$$T = 8 \max \left\{ \frac{\mathscr{T}}{3}, \frac{(f(x_0) - f^*)\mathscr{T}}{\mathscr{F}}, \frac{f(x_0) - f^*}{\eta\epsilon^2} \right\}. \tag{12}$$

PRGD performs two types of steps: (1) if $\|\operatorname{grad} f(x_t)\| > \epsilon$, an RGD step on the manifold, and (2) if $\|\operatorname{grad} f(x_t)\| \leq \epsilon$, a perturbation in the tangent space followed by GD steps in the tangent space.

There are at most $T/4$ iterates $x_t \in \mathcal{M}$ satisfying $\|\operatorname{grad} f(x_t)\| > \epsilon$ (i.e., iterates where an RGD step is performed), for otherwise Lemma B.1 and the definition of $T$ (12) would imply $f(x_T) < f(x_0) - T\eta\epsilon^2/8 \leq f^*$, which contradicts Assumption 1.

The variable $t$ in Algorithm 1 is an upper bound on the number of gradient queries issued so far. For each RGD step on the manifold, $t$ increases by exactly 1. PRGD does not terminate before $t$ exceeds $T$, and for every perturbation the counter increases by exactly $\mathscr{T}$. Therefore, there are at least $3T/(4\mathscr{T})$ iterates $x_t \in \mathcal{M}$ satisfying $\|\operatorname{grad} f(x_t)\| \leq \epsilon$. By the definition of $T$ (12), $3T/(4\mathscr{T}) \geq 2$.

Suppose PRGD visits more than $T/(4\mathscr{T})$ points $x_t \in \mathcal{M}$ satisfying $\|\operatorname{grad} f(x_t)\| \leq \epsilon$ and $\lambda_{\min}(\nabla^2 \hat{f}_{x_t}(0)) \leq -\sqrt{\rho\epsilon}$. Each of these iterates $x_t$ is followed by a perturbation and at most $\mathscr{T}$ tangent space steps $\{s_j\}$. For at least one such $x_t$, the sequence of tangent space steps does not escape the saddle point (that is, $f(x_{t+\mathscr{T}}) - f(x_t) > -\mathscr{F}/2$), for otherwise $f(x_T) < f(x_0) - T\mathscr{F}/(8\mathscr{T}) \leq f^*$ by the definition of $T$ (12). Yet, by Lemma B.2 and a union bound, the probability that one or more of these sequences does not escape is at most $\delta$. Indeed, factoring out the third term in the max,

$$T = \frac{8\ell(f(x_0) - f^*)}{\epsilon^2} \max \left\{ \frac{1}{3} \frac{\chi}{\sqrt{\rho\epsilon}} \frac{\epsilon^2}{(f(x_0) - f^*)}, 50\chi^4, 1 \right\}$$

$$\leq \frac{8\ell(f(x_0) - f^*)}{\epsilon^2} \max \left\{ \chi, 50\chi^4, 1 \right\} = O\left( \frac{\ell(f(x_0) - f^*)}{\epsilon^2} \chi^4 \right),$$

where we used $\epsilon^{3/2} \leq 3\sqrt{\rho}\,(f(x_0) - f^*)$. Now using

$$\max \left\{ \chi, 50\chi^4, 1 \right\} \leq 2^{18 + \chi/4}$$

for all $\chi > 1/4$, and $\chi \geq 4\log_2 \left( 2^{31} \frac{\ell^2 \sqrt{d}(f(x_0) - f^*)}{\delta\sqrt{\rho}\epsilon^{5/2}} \right)$, we find

$$T \cdot \frac{\ell\sqrt{d}}{\sqrt{\rho\epsilon}} 2^{10 - \chi/2} \leq \frac{\ell^2 \sqrt{d}}{\sqrt{\rho\epsilon}} \frac{(f(x_0) - f^*)}{\epsilon^2} 2^{31 - \chi/4} \leq \delta,$$

as announced.

Hence, with probability at least $1 - \delta$, PRGD visits at most $T/(4\mathscr{T})$ points $x_t$ satisfying $\|\operatorname{grad} f(x_t)\| \leq \epsilon$ and $\lambda_{\min}(\nabla^2 \hat{f}_{x_t}(0)) \leq -\sqrt{\rho\epsilon}$. Using that there are at least $3T/(4\mathscr{T})$ iterates $x_t \in \mathcal{M}$ with $\|\operatorname{grad} f(x_t)\| \leq \epsilon$, we conclude that at least two-thirds of the iterates $x_t \in \mathcal{M}$ with $\|\operatorname{grad} f(x_t)\| \leq \epsilon$ also satisfy $\lambda_{\min}(\nabla^2 \hat{f}_{x_t}(0)) \geq -\sqrt{\rho\epsilon}$, with probability at least $1 - \delta$. $\square$

Corollary 3.5 follows directly from Theorem 3.4 and the following lemma.

**Lemma B.3.** *For some $\rho > 0$ (which would typically come from Assumption 3), under Assumption 4 on the retraction, let $x \in \mathcal{M}$ satisfy $\|\operatorname{grad} f(x)\| \leq \epsilon$ and $\lambda_{\min}(\nabla^2 \hat{f}_x(0)) \geq -\sqrt{\rho\epsilon}$. Then, $\lambda_{\min}(\operatorname{Hess} f(x)) \geq -\sqrt{\rho\epsilon} - \beta\epsilon$. In particular, if $\epsilon \leq \rho/\beta^2$, then $\lambda_{\min}(\operatorname{Hess} f(x)) \geq -\sqrt{4\rho\epsilon}$.*

*Proof.* Considering $s = 0$ in Lemma 2.2, we may use $\operatorname{Retr}_x(0) = x$ and that $T_{x,0}$ is the identity (as per Definition 2.1) to get $\nabla^2 \hat{f}_x(0) = \operatorname{Hess} f(x) + W_0$, where

$$\forall \dot{s} \in \mathrm{T}_x\mathcal{M} \text{ with } \|\dot{s}\| = 1, \qquad \langle W_0[\dot{s}], \dot{s} \rangle \leq \left\| \gamma''_{x,\dot{s}}(0) \right\| \|\operatorname{grad} f(x)\| \leq \beta\epsilon.$$

Thus, $\|W_0\| \leq \beta\epsilon$ and we find $\lambda_{\min}(\operatorname{Hess} f(x)) \geq -\sqrt{\rho\epsilon} - \beta\epsilon$. For the last part, use $\beta \leq \sqrt{\rho/\epsilon}$. $\square$

Corollary 3.6 follows directly from Corollary 3.5 and Lemma 3.2.

# C  Proofs of key lemmas

The goal of this section is to prove Lemmas B.1 and B.2. All proofs deal with linear spaces, not manifolds. The key ideas are due to Jin et al. [18]. The following lemma is needed because to apply Jin et al.'s analysis we need the pullbacks not only to satisfy the restricted Lipschitz condition, Assumption 2, but also to have Lipschitz continuous gradient at least, uniformly in tangent space balls of fixed radius. The lemma below implies Lemma 3.3.

**Lemma C.1.** *Let $f$ satisfy Assumptions 2 and 3, and let $\ell = L + \rho b$. For all $x \in \mathcal{M}$, it holds that $\nabla \hat{f}_x$ is $\ell$-Lipschitz continuous in the ball $B_{x,b}(0) \subset \mathrm{T}_x\mathcal{M}$.*

*Proof.* By Assumption 2, $\left\| \nabla^2 \hat{f}_x(0) \right\| \leq L$. Hence, by Assumption 3, for all $s \in B_{x,b}(0)$,

$$\left\| \nabla^2 \hat{f}_x(s) \right\| \leq \left\| \nabla^2 \hat{f}_x(0) \right\| + \left\| \nabla^2 \hat{f}_x(s) - \nabla^2 \hat{f}_x(0) \right\| \leq L + \rho \left\| s \right\| \leq L + \rho b = \ell.$$

Let $s_1, s_2 \in B_{x,b}(0)$ be arbitrary. Then indeed,

$$\left\| \nabla \hat{f}_x(s_2) - \nabla \hat{f}_x(s_1) \right\| = \left\| \int_0^1 \nabla^2 \hat{f}_x(s_1 + (s_2 - s_1)\tau)[s_2 - s_1] d\tau \right\| \leq \ell \left\| s_2 - s_1 \right\|, \quad (13)$$

where we used that the line segment from $s_1$ to $s_2$ is contained in $B_{x,b}(0)$. $\qquad\square$

Together with the one above, the following standard lemma allows us to establish the sufficient decrease of $\hat{f}_x$ in $B_{x,b}(0)$ upon taking a gradient step in the tangent space.

**Lemma C.2.** *Let $\nabla \hat{f}_x$ be $\ell$-Lipschitz continuous along the line segment connecting $s_j$ to $s_{j+1}$, related by $s_{j+1} = s_j - \alpha\eta\nabla\hat{f}_x(s_j)$ with $\eta = 1/\ell$ and $\alpha \in [0,1]$. Then,*

$$\hat{f}_x(s_{j+1}) - \hat{f}_x(s_j) \leq -\frac{\alpha\eta}{2} \left\| \nabla\hat{f}_x(s_j) \right\|^2.$$

*Proof.* It is a standard consequence of Lipschitz continuity of $\nabla\hat{f}_x$ along the line segment $\tau \mapsto (1 - \tau)s_j + \tau s_{j+1}$ for $\tau \in [0,1]$ that

$$\hat{f}_x(s_{j+1}) \leq \hat{f}_x(s_j) + \left\langle \nabla\hat{f}_x(s_j), s_{j+1} - s_j \right\rangle + \frac{\ell}{2} \left\| s_{j+1} - s_j \right\|^2.$$

Plugging in $s_{j+1} - s_j = -\alpha\eta\nabla\hat{f}_x(s_j)$, we get

$$\hat{f}_x(s_{j+1}) \leq \hat{f}_x(s_j) + \left[ -\alpha\eta + \frac{\ell\alpha^2\eta^2}{2} \right] \left\| \nabla\hat{f}_x(s_j) \right\|^2.$$

The coefficient between brackets is further equal to $\left( -1 + \frac{\alpha}{2} \right) \alpha\eta$, which is at most $-\alpha\eta/2$. $\qquad\square$

We are now ready to prove Lemma B.1.

*Proof of Lemma B.1.* The call to TANGENTSPACESTEPS$(x, 0, \eta, b, 1)$ produces a point $\mathrm{Retr}_x(s_1)$, with $s_1 = s_0 - \alpha\eta\nabla\hat{f}_x(s_0)$, where $s_0 = 0$, $\alpha \in [0,1]$ and $\|s_1\| \leq b$. Owing to Assumption 2, we know that $\nabla\hat{f}_x$ is $L$-Lipschitz continuous along the line segment connecting $s_0$ to $s_1$. Since $\ell \geq L$, it is a fortiori $\ell$-Lipschitz continuous along that line segment: Lemma C.2 applies and yields

$$f(\mathrm{Retr}_x(s_1)) = \hat{f}_x(s_1) \leq \hat{f}_x(s_0) - \frac{\alpha\eta}{2} \left\| \nabla\hat{f}_x(s_0) \right\|^2 = f(x) - \frac{\alpha\eta}{2} \left\| \nabla\hat{f}_x(0) \right\|^2.$$

If $\alpha = 1$, since $\left\| \nabla\hat{f}_x(0) \right\| = \|\mathrm{grad}\, f(x)\| > \epsilon$, we are done. Owing to how TANGENTSPACESTEPS works, if $\alpha < 1$, then it must be that $\|\alpha\eta\nabla\hat{f}_x(0)\| = b$, so that the inequality above yields

$$f(\mathrm{Retr}_x(s_1)) \leq f(x) - \frac{b}{2} \left\| \nabla\hat{f}_x(0) \right\| \leq f(x) - \frac{b\epsilon}{2}.$$

Using $\epsilon \leq b^2 \rho$ and $\ell \geq L \geq \sqrt{\rho\epsilon}$,

$$\eta\epsilon = \frac{\epsilon}{\ell} \leq \frac{\sqrt{b^2\rho\epsilon}}{\ell} = \frac{\sqrt{\rho\epsilon}}{\ell} b \leq b.$$

Hence, $f(\mathrm{Retr}_x(s_1)) \leq f(x) - \eta\epsilon^2/2$, as desired. (As a side note: Assumption 3 is not truly necessary here; it is only convenient so that we can use the same definitions of $\rho, b$ and $\ell$ as in other parts of the paper.) $\qquad\square$

Lemma C.3 is Jin et al.'s "improve or localize lemma" [18], with a tweak for variable step sizes. The lemma states that if the function value does not decrease much, then the iterates are localized.

**Lemma C.3.** *Fix $j \geq 0$, $x \in \mathcal{M}$ and $s_0 \in \mathrm{T}_x\mathcal{M}$. For all $0 \leq i \leq j-1$, assume $0 \leq \eta_i \leq \eta = 1/\ell$, $s_{i+1} = s_i - \eta_i \nabla \hat{f}_x(s_i)$ and $\nabla \hat{f}_x$ is $\ell$-Lipschitz continuous along the line segment connecting $s_i$ to $s_{i+1}$. Then,*

$$\|s_j - s_0\| \leq \sqrt{2\eta j (\hat{f}_x(s_0) - \hat{f}_x(s_j))}.$$

*Proof.* Using a telescoping sum, triangle inequality, Cauchy–Schwarz and (to get to the last line) Lemma C.2, we get:

$$\|s_j - s_0\| = \left\| \sum_{i=0}^{j-1} s_{i+1} - s_i \right\| = \left\| \sum_{i=0}^{j-1} -\eta_i \nabla \hat{f}_x(s_i) \right\| \leq \sum_{i=0}^{j-1} \sqrt{\eta_i} \left\| \sqrt{\eta_i} \nabla \hat{f}_x(s_i) \right\|$$

$$\leq \sqrt{\left( \sum_{i=0}^{j-1} \eta_i \left\| \nabla \hat{f}_x(s_i) \right\|^2 \right) \left( \sum_{i=0}^{j-1} \eta_i \right)} \leq \sqrt{2\eta j \left( \sum_{i=0}^{j-1} \frac{\eta_i}{2} \left\| \nabla \hat{f}_x(s_i) \right\|^2 \right)}$$

$$\leq \sqrt{2\eta j \left( \sum_{i=0}^{j-1} \hat{f}_x(s_i) - \hat{f}_x(s_{i+1}) \right)} = \sqrt{2\eta j (\hat{f}_x(s_0) - \hat{f}_x(s_j))}. \qquad\square$$

Lemma C.4 below and its proof are very similar to Jin et al.'s Lemma 13 and its proof [18], except for a modification since $\nabla \hat{f}_x$ is only Lipschitz continuous in a ball. This deterministic lemma formalizes the coupling sequence argument: if the Hessian of the pullback has a negative eigenvalue which is large in magnitude, upon initializing the tangent space steps at two appropriately chosen points $s_0, s_0'$, with certainty, one of them leads to significant decrease in the cost function. As usual, we use parameters $\eta, r, \mathscr{T}$ as in (9) and $\mathscr{F}, \mathscr{L}$ as in (10).

**Lemma C.4.** *Under Assumptions 2 and 3, let $x \in \mathcal{M}$ be such that $\lambda_{\min}(\nabla^2 \hat{f}_x(0)) \leq -\sqrt{\rho\epsilon}$, with $\epsilon \leq b^2 \rho$ and $L \geq \sqrt{\rho\epsilon}$. Let $s_0, s_0' \in \mathrm{T}_x\mathcal{M}$ be such that*

1. *$\|s_0\|, \|s_0'\| \leq \eta r$, and*

2. *$s_0 - s_0' = \eta r_0 e_1$, where $e_1$ is an eigenvector of unit norm associated with the minimum eigenvalue of $\nabla^2 \hat{f}_x(0)$, and $r_0 > \omega = 2^{2-\chi}\ell\mathscr{L}$.*

*Let $s_{\mathscr{T}}$ be defined by running* TANGENTSPACESTEPS$(x, s_0, \eta, b, \mathscr{T})$ *(see Algorithm 1). Let $s'_{\mathscr{T}}$ be similarly defined by running* TANGENTSPACESTEPS$(x, s_0', \eta, b, \mathscr{T})$. *Then,*

$$\min\left\{ \hat{f}_x(s_{\mathscr{T}}) - \hat{f}_x(s_0), \hat{f}_x(s'_{\mathscr{T}}) - \hat{f}_x(s_0') \right\} \leq -\mathscr{F}.$$

*Proof.* First, note that both sequences are initialized in the interior of the ball of radius $b$. Indeed, using $\ell \geq L, L \geq \sqrt{\rho\epsilon}, \epsilon \leq b^2 \rho$ and $\chi > 1/4$,

$$\eta r = \frac{1}{\ell} \frac{\epsilon}{400\chi^3} < \frac{\epsilon}{L} \frac{64}{400} = b\sqrt{\frac{\rho\epsilon}{L^2} \frac{\epsilon}{b^2\rho}} \frac{64}{100} \leq b\frac{64}{100} < b. \qquad (14)$$

The proof is by contradiction: assume

$$\min\left\{ \hat{f}_x(s_{\mathscr{T}}) - \hat{f}_x(s_0), \hat{f}_x(s'_{\mathscr{T}}) - \hat{f}_x(s_0') \right\} > -\mathscr{F}.$$

Further assume, for the sake of contradiction, that one of the sequences $\{s_j\}_{j\leq\mathscr{T}}, \{s_j'\}_{j\leq\mathscr{T}}$ (defined in TANGENTSPACESTEPS) escapes the interior of the ball $B_{x,b}(0)$. Without loss of generality, assume $\{s_j\}_{j\leq\mathscr{T}}$ escapes. Let $j \leq \mathscr{T} - 1$ be the minimum integer for which $\|s_{j+1}\| \geq b$. Then, TANGENTSPACESTEPS$(x, s_0, \eta, b, \mathscr{T})$ terminates with $s_j - \alpha\eta\nabla\hat{f}_x(s_j)$ for some $\alpha \in (0,1]$ satisfying $b = \left\|s_j - \alpha\eta\nabla\hat{f}_x(s_j)\right\|$. Using Lemma C.3, $\ell \geq L \geq \sqrt{\rho\epsilon}$ and $\chi > \frac{1}{4}$,

$$b = \left\|s_j - \alpha\eta\nabla\hat{f}_x(s_j)\right\| \leq \left\|s_j - \alpha\eta\nabla\hat{f}_x(s_j) - s_0\right\| + \|s_0\| \leq \sqrt{2\eta(j+1)\mathscr{F}} + \eta r$$

$$\leq \sqrt{2\eta\mathscr{T}\mathscr{F}} + \eta r \leq \sqrt{\frac{\epsilon}{25\chi^2\rho}} + \frac{1}{400\chi^3}\sqrt{\frac{\epsilon}{\rho}} \leq \left(\frac{1}{5\chi} + \frac{1}{400\chi^3}\right)\sqrt{\frac{\epsilon}{\rho}} \leq \frac{1}{4\chi}\sqrt{\frac{\epsilon}{\rho}} = \mathscr{L}.$$

Since $\epsilon \leq b^2\rho$, we know that $\mathscr{L} < b$, which shows a contradiction. Thus, neither of the sequences $\{s_j\}_{j\leq\mathscr{T}}, \{s_j'\}_{j\leq\mathscr{T}}$ leave the interior of $B_{x,b}(0)$. That is, $s_{j+1} = s_j - \eta\nabla\hat{f}_x(s_j)$ and $\|s_{j+1}\| < b$ for $j = 0, 1, \ldots, \mathscr{T} - 1$, and similarly for $\{s_j'\}_{j\leq\mathscr{T}}$.

From here, we proceed exactly as in Lemma 13 of [18]. By Lemma C.3, for all $j \leq \mathscr{T}$,

$$\max\left\{\|s_j\|, \|s_j'\|\right\} \leq \max\left\{\|s_j - s_0\|, \|s_j' - s_0'\|\right\} + \eta r \leq \sqrt{2\eta\mathscr{T}\mathscr{F}} + \eta r \leq \mathscr{L}. \tag{15}$$

Let $\hat{s}_j = s_j - s_j'$ and $\mathcal{H} = \nabla^2\hat{f}_x(0)$. Then,

$$\hat{s}_{j+1} = \hat{s}_j - \left(\eta\nabla\hat{f}_x(s_j) - \eta\nabla\hat{f}_x(s_j')\right) = \hat{s}_j - \eta\int_0^1 \nabla^2\hat{f}_x\left(s_j' + \theta(s_j - s_j')\right)[s_j - s_j']d\theta$$

$$= (I - \eta\mathcal{H})\hat{s}_j - \eta\Delta_j\hat{s}_j,$$

where $\Delta_j = \int_0^1\left(\nabla^2\hat{f}_x\left(s_j' + \theta(s_j - s_j')\right) - \mathcal{H}\right)d\theta$. By Assumption 3,

$$\|\Delta_j\| \leq \int_0^1 \rho\left\|s_j' + \theta(s_j - s_j')\right\|d\theta \leq \int_0^1 \rho\max\{\|s_j\|, \|s_j'\|\}d\theta \leq \rho\mathscr{L}.$$

This will be useful momentarily. It is easy to check by induction that

$$\hat{s}_{j+1} = p(j+1) - q(j+1),$$

where $p(0) = \hat{s}_0, q(0) = 0$, and

$$p(j+1) = (I - \eta\mathcal{H})^{j+1}\hat{s}_0, \qquad \text{and} \qquad q(j+1) = \eta\sum_{i=0}^{j}(I - \eta\mathcal{H})^{j-i}\Delta_i\hat{s}_i.$$

We use induction to show that $\|q(j)\| \leq \|p(j)\|/2$. The claim is clearly true for $j = 0$. Suppose the claim is true for all $i \leq j$. We prove the claim for $j + 1$. Let $-\gamma = \lambda_{\min}(\nabla^2\hat{f}_x(0))$. Using $\hat{s}_0 = \eta r_0 e_1$, notice in particular that

$$p(j) = (I - \eta\mathcal{H})^j\eta r_0 e_1 = (1 + \eta\gamma)^j\eta r_0 e_1,$$

so that the norm of $p(j)$ grows with $j$: $\|p(j)\| = (1 + \eta\gamma)^j\eta r_0$. Using the induction hypothesis, for all $i \leq j$ we have:

$$\|\hat{s}_i\| \leq \|p(i)\| + \|q(i)\| \leq \frac{3}{2}\|p(i)\| \leq 2(1 + \eta\gamma)^i\eta r_0.$$

Furthermore, since $\mathcal{H} \preceq LI \preceq \ell I$, it follows that $I - \eta\mathcal{H} \succeq 0$. As a result, $\|I - \eta\mathcal{H}\| = \lambda_{\max}(I - \eta\mathcal{H}) = 1 + \eta\gamma$. Therefore, also using $2\eta\rho\mathscr{L}\mathscr{T} = 1/2$ in the last step,

$$\|q(j+1)\| = \left\|\eta\sum_{i=0}^{j}(I - \eta\mathcal{H})^{j-i}\Delta_i\hat{s}_i\right\| \leq \eta\rho\mathscr{L}\sum_{i=0}^{j}\left\|(I - \eta\mathcal{H})^{j-i}\right\|\|\hat{s}_i\|$$

$$\leq 2\eta\rho\mathscr{L}\sum_{i=0}^{j}(1 + \eta\gamma)^{j-i}(1 + \eta\gamma)^i\eta r_0$$

$$\leq 2\eta\rho\mathscr{L}\mathscr{T}(1 + \eta\gamma)^j\eta r_0 = 2\eta\rho\mathscr{L}\mathscr{T}\|p(j)\| \leq \|p(j+1)\|/2,$$

So we have proven $\|q(j)\| \le \|p(j)\| / 2$ for all $j$. Therefore, using the definition of $r_0$ in the last step,

$$\max\{\|s_\mathscr{T}\|, \|s'_\mathscr{T}\|\} \ge (\|s_\mathscr{T}\| + \|s'_\mathscr{T}\|)/2 \ge \|\hat{s}_\mathscr{T}\|/2 \ge (\|p(\mathscr{T})\| - \|q(\mathscr{T})\|)/2$$
$$\ge \|p(\mathscr{T})\|/4 = (1+\eta\gamma)^\mathscr{T}\eta r_0/4 \ge 2^{\chi-2}\eta r_0 > \mathscr{L},$$

which contradicts (15). In the second to last step, we used $\gamma \ge \sqrt{\rho\epsilon}$ and $\sqrt{\rho\epsilon} \le \ell$ so that

$$\frac{1}{\chi}\log_2\left((1+\eta\gamma)^\mathscr{T}\right) \ge \frac{\mathscr{T}}{\chi}\log_2\left(1 + \frac{\sqrt{\rho\epsilon}}{\ell}\right) = \frac{\mathscr{T}}{\chi}\log_2\left(1 + \frac{\chi}{\mathscr{T}}\right) \ge 1,$$

since $\frac{1}{\alpha}\log_2(1+\alpha) \ge 1$ for all $\alpha \in [0,1]$. Except for the initial part, this proof is due to Jin et al. [18]. $\qquad\square$

We are now ready to prove Lemma B.2. This proof is completely due to Jin et al. [18]: we only somewhat modify how the proof is presented.

*Proof of Lemma B.2.* Recall that $\eta r < b$ (14), and define the stuck region

$$\mathcal{X}_{\text{stuck}} = \big\{s \in B_{x,\eta r}(0) : f(\text{TANGENTSPACESTEPS}(x, s, \eta, b, \mathscr{T})) - f(x) > -\mathscr{F}\big\}.$$

Running the tangent space steps with $s_0$ in that set does not yield sufficient improvement of the cost function despite the fact that the Hessian has a negative eigenvalue with large magnitude, hence the name. We aim to show that this set has a small volume, so that it is unlikely to encounter it by random chance.

Let $S_{e_1}$ be the subspace of $\mathrm{T}_x\mathcal{M}$ orthogonal to $e_1$. Given $a \in S_{e_1} \cap B_{x,\eta r}(0)$, let $\ell_a$ denote the line in $\mathrm{T}_x\mathcal{M}$ parallel to $e_1$ passing through $a$. Then, with $\mathbb{1}$ denoting the indicator function,

$$\mathrm{Vol}(\mathcal{X}_{\text{stuck}}) = \int_{\mathrm{T}_x\mathcal{M}} \mathbb{1}_{\mathcal{X}_{\text{stuck}}}(y)dy = \int_{S_{e_1}\cap B_{x,\eta r}(0)}\left[\int_{\ell_a}\mathbb{1}_{\mathcal{X}_{\text{stuck}}}(z)dz\right]da.$$

Lemma C.4 states any two points that are both on the line $\ell_a$ *and* in $\mathcal{X}_{\text{stuck}}$ must be close. Specifically, for all $s, s' \in \ell_a \cap \mathcal{X}_{\text{stuck}}$, we have $\|s - s'\| \le \eta\omega$, with $\omega = 2^{2-\chi}\ell\mathscr{L}$. Therefore, the set of problematic points on the line $\ell_a$ is contained in a segment of length at most $\eta\omega$ and we deduce $\int_{\ell_a}\mathbb{1}_{\mathcal{X}_{\text{stuck}}}(z)dz \le \eta\omega$. As a result,

$$\mathrm{Vol}(\mathcal{X}_{\text{stuck}}) \le \eta\omega\int_{S_{e_1}\cap B_{x,\eta r}(0)} da = \eta\omega\mathrm{Vol}(\mathbb{B}_{\eta r}^{d-1}),$$

where $\mathbb{B}_R^k$ denotes a $k$-dimensional (Euclidean) ball of radius $R$. Since $s_0 \sim \mathrm{Uniform}(B_{x,\eta r}(0))$,

$$\mathbb{P}(s_0 \in \mathcal{X}_{\text{stuck}}) = \frac{\mathrm{Vol}(\mathcal{X}_{\text{stuck}})}{\mathrm{Vol}(\mathbb{B}_{\eta r}^d)} \le \frac{\eta\omega\mathrm{Vol}(\mathbb{B}_{\eta r}^{d-1})}{\mathrm{Vol}(\mathbb{B}_{\eta r}^d)} = \frac{\omega\Gamma(1+d/2)}{r\sqrt{\pi}\Gamma((d+1)/2)} \le \frac{\omega}{r}\sqrt{\frac{d}{\pi}} = \frac{\ell\sqrt{d}}{\sqrt{\rho\epsilon}}\frac{400}{\sqrt{\pi}}2^{-\chi}\chi^2$$

$$\le \frac{\ell\sqrt{d}}{\sqrt{\rho\epsilon}}2^{10-\chi/2},$$

where we used the Gautschi inequality for the $\Gamma$ function, and $\chi > 1/4$ to bound $\frac{400}{\sqrt{\pi}}2^{-\chi}\chi^2 \le 2^{10-\chi/2}$. To conclude, note that if $s_0 \notin \mathcal{X}_{\text{stuck}}$ then

$$f(\text{TANGENTSPACESTEPS}(x, s_0, \eta, b, \mathscr{T})) - f(x)$$
$$= f(\text{TANGENTSPACESTEPS}(x, s_0, \eta, b, \mathscr{T})) - \hat{f}_x(s_0) + \hat{f}_x(s_0) - \hat{f}_x(0)$$
$$\le -\mathscr{F} + \epsilon\eta r + \ell\eta^2 r^2/2 = -\mathscr{F} + \frac{\sqrt{\rho\epsilon}}{\ell}\mathscr{F}\left(50\chi^3\right)\left(\frac{1}{400\chi^3} + \frac{1}{2}\left(\frac{1}{400\chi^3}\right)^2\right)$$
$$\le -\mathscr{F} + \mathscr{F}\left(\frac{1}{8} + \frac{25}{400^2\chi^3}\right) \le -\mathscr{F}/2,$$

using $\sqrt{\rho\epsilon} \le \ell$ and $\chi > 1/4$ once more in the last step, and also

$$\hat{f}_x(s_0) - \hat{f}_x(0) \le \epsilon\eta r + \ell\eta^2 r^2/2$$

owing to the fact that $\nabla\hat{f}_x$ is $\ell$-Lipschitz continuous along the line segment connecting $0$ and $s_0$, $\|s_0\| \le \eta r$ and $\|\nabla\hat{f}_x(0)\| \le \epsilon$. $\qquad\square$

# D   Regularity constants for dominant eigenvector computation (PCA)

Computing the dominant eigenvector of a symmetric matrix $A \in \mathbb{R}^{n \times n}$ (which notably comes up in PCA) comes down to solving

$$\max_{x \in S^{n-1}} f(x), \qquad\qquad f(x) = \frac{1}{2} x^\top A x, \tag{16}$$

where $S^{n-1} = \{x \in \mathbb{R}^n : x^\top x = 1\}$ is the unit sphere. If we use the retraction $\mathrm{Retr}_x(s) = \frac{x+s}{\|x+s\|}$ (where $\|x\| = \sqrt{x^\top x}$)—for which Assumption 4 holds with $\beta = 0$—then pullbacks are of the form

$$\hat{f}_x(s) = f(\mathrm{Retr}_x(s)) = \frac{1}{1 + \|s\|^2} \frac{1}{2} (x+s)^\top A(x+s), \tag{17}$$

defined over the tangent spaces $\mathrm{T}_x S^{n-1} = \{s \in \mathbb{R}^n : x^\top s = 0\}$. The gradient of $\hat{f}_x$ at $s$ is given by

$$\nabla \hat{f}_x(s) = \mathrm{Proj}_x\left( \frac{1}{1 + \|s\|^2} A(x+s) + \frac{-1}{(1 + \|s\|^2)^2}(x+s)^\top A(x+s) \cdot s \right) \tag{18}$$

$$= \frac{1}{1 + \|s\|^2}\left( \mathrm{Proj}_x(A(x+s)) - 2\hat{f}_x(s) \cdot s \right), \tag{19}$$

where $\mathrm{Proj}_x(s) = s - (x^\top s)x$ is the orthogonal projector from $\mathbb{R}^n$ to $\mathrm{T}_x S^{n-1}$. It follows that

$$\nabla \hat{f}_x(s) - \nabla \hat{f}_x(0) = \frac{1}{1 + \|s\|^2}\left( \mathrm{Proj}_x(As) - 2\hat{f}_x(s)s - \|s\|^2 \mathrm{Proj}_x(Ax) \right). \tag{20}$$

Using $\frac{1}{1 + \|s\|^2} \leq 1$ and $\frac{\|s\|^2}{1 + \|s\|^2} \leq \frac{1}{2}\|s\|$ for all $s$, and using the fact that an orthogonal projector can only reduce the norm of a vector, we find

$$\|\nabla \hat{f}_x(s) - \nabla \hat{f}_x(0)\| \leq \|As\| + 2\left[ \sup_{s \in \mathrm{T}_x S^{n-1}} |\hat{f}_x(s)| \right] \|s\| + \frac{1}{2}\|Ax\|\|s\|. \tag{21}$$

Letting $\|A\|$ denote the operator norm of $A$ (largest singular value), we finally obtain

$$\|\nabla \hat{f}_x(s) - \nabla \hat{f}_x(0)\| \leq \frac{5}{2}\|A\|\|s\|. \tag{22}$$

This shows that Assumption 2 holds with $b_1 = \infty$ and $L = \frac{5}{2}\|A\|$, or any larger number. For example, the induced 1-norm of the matrix $A$ is straightforward to compute and is an upper-bound on $\|A\|$.

Now aiming to control second-order derivatives, we compute a directional derivative of $\nabla \hat{f}_x(s)$ and obtain the Hessian of $\hat{f}_x$ on the tangent space $\mathrm{T}_x S^{n-1}$:

$$\nabla^2 \hat{f}_x(s)[\dot{s}] = -2\frac{\langle s, \dot{s}\rangle}{1 + \|s\|^2}\nabla \hat{f}_x(s) + \frac{1}{1 + \|s\|^2}\left[ \mathrm{Proj}_x(A\dot{s}) - 2\hat{f}_x(s)\dot{s} - 2\langle \nabla \hat{f}_x(s), \dot{s}\rangle s \right],$$

where $\langle u, v\rangle = u^\top v$. In particular, $\nabla^2 \hat{f}_x(0)[\dot{s}] = \mathrm{Proj}_x(A\dot{s}) - (x^\top A x)\dot{s}$, so that

$$\left\langle \dot{s}, \left( \nabla^2 \hat{f}_x(s) - \nabla^2 \hat{f}_x(0) \right)[\dot{s}] \right\rangle = -4\frac{\langle s, \dot{s}\rangle \langle \nabla \hat{f}_x(s), \dot{s}\rangle}{1 + \|s\|^2} + \left( \frac{1}{1 + \|s\|^2} - 1 \right)\langle \dot{s}, A\dot{s}\rangle$$

$$- \left( 2\frac{\hat{f}_x(s)}{1 + \|s\|^2} - x^\top A x \right)\|\dot{s}\|^2. \tag{23}$$

Using $\frac{\|s\|}{1 + \|s\|^2} \leq \frac{1}{2}$, it is easy to see that $\|\nabla \hat{f}_x(s)\| \leq \frac{3}{2}\|A\|$ and:

$$\|\nabla^2 \hat{f}_x(s) - \nabla^2 \hat{f}_x(0)\| \leq 4\|\nabla \hat{f}_x(s)\|\|s\| + \frac{1}{2}\|A\|\|s\|$$

$$+ \left[ \sup_{s \in \mathrm{T}_x S^{n-1}} \frac{\left| (x+s)^\top A(x+s) - (1 + \|s\|^2)^2 x^\top A x \right|}{(1 + \|s\|^2)^2 \|s\|} \right] \|s\|$$

$$\leq (6 + 1/2)\|A\|\|s\| + \left[ \sup_{t > 0} \frac{2t + 3t^2 + t^4}{(1 + t^2)^2 t} \right]\|A\|\|s\|$$

$$\leq 9\|A\|\|s\|.$$

This shows Assumption 3 holds with $b_2 = \infty$ and $\rho = 9\|A\|$.