[Reviews · NeurIPS 2019]

Reviewer 1



========================================================== After reading response, I do agree that getting the clean results and analysis is even nicer (compared to complicated analysis). It is also an important contribution to get the results for escaping saddle points for manifold. I will vote for acceptance. ========================================================== The result is very nice. It's only slightly unsatisfactory that smooth assumption 3.2 and 3.3 are in terms of function dot retractions, which make these assumptions a bit indirect and hard to verify. It also hides interesting information about the geometry of manifold there.

Reviewer 2



This paper is a nice extension on Jin et al's work. However, I have two concerns. First, the analysis is very similar to Jin et al's except a few steps that are needed to connect the gradient/Hessian of the original function and those of the pullback function. This reduces the novelty of this paper in its technical analysis a little bit. Second, I am not sure how b can be estimated. Although in practice, many parameters can be tuned but I am not sure where to start when I have to tune b. Although in the examples in the paper b=+infinity, I believe there are cases where b is finite and unknown. For other parameters such as L and rho, we can at least give some estimation (may be conservative) but, for b, it is not clear how. Are there any examples where b<+infty and some conservative estimation of b are possible? ============================================== I have read the authors' response and I am satisfied with that. I will slightly increase my score.

Reviewer 3



This paper shows that a perturbed variant of gradient descent on Riemannian manifolds can also escape saddle points efficiently. The assumptions and the preliminaries are clearly described in the paper. The PRGD algorithm for Riemannian manifolds (analogous to PRGD on Euclidean spaces) involves taking one of the two steps: If the gradient is large, then perform gradient descent else perturb and take tau gradient steps along the tangent space. The authors should further discuss setting step-size, perturbation radius and no of perturbation steps in the context of different examples to highlight the practicality of their algorithm. Post rebuttal: Having read the author's responses, I would like to keep my current score.

Reviewer 4



This paper proposes a perturbed Riemannian gradient descent method (PRGD) for minimizing a smooth and nonconvex function over Riemannian manifold. It sheds light on understanding RGD and its capability for escaping from saddle points. There are two main concerns about this work. 1. The algorithm depends on many unknown parameters which make it impractical, although other works such as Jin et.al. in Euclidean setting have the same issue. 2. This is more serious. The algorithm design and proof techniques are largely the same as the ones in previous works by Jin et.al. The same topic has also been studied by Sun et.al. on submanifolds of Euclidean space. As a result, it seems that the only contribution of this paper is to formalize or generalize these techniques to more general manifolds. Therefore, the contribution is a bit thin. ----------------------------------------------- Comments after rebuttal: I appreciate the authors' detailed feedback, which clarified some of my previous concerns. I thus increased my score.

[Author Response · NeurIPS 2019]

We thank the reviewers for their time and observations. The main points raised are:

1. Our analysis is close to Jin et al.'s, limiting the contribution.

2. Regularity assumptions pertain to $f \circ R_x$ rather than to $f$ directly.

3. The algorithm parameters are difficult to determine.

Regarding the first point, we argue that getting the analysis to generalize in a simple way is precisely the contribution. To
support the claim that this is not direct, we ask the reviewers to consider the increase in abstraction level and technicality
separating Jin et al.'s work from Sun and Fazel's first paper, and also from the more recent work by Sun, Flammarion
and Fazel on the same topic (appeared on arXiv after the submission deadline), and to compare this increase to the
one separating Jin et al.'s work from our paper. Our more direct generalization, based on a non-standard separation of
manifold and tangent steps and on a particular adaptation of the regularity conditions, does not require Riemannian
distances, exponentials, logarithms or curvature: all sophisticated objects that require careful considerations (not always
spelled out in the existing literature). Yet, our approach yields a more general statement, closer to the original by Jin et
al., and, eventually, with less friction. Because of this, there is a better chance that this approach may serve in other
contexts as well (we are confident it can extend to stochastic gradients with some work), and that it will be used by
others. In our opinion, this is part of the role of a theory-inclined paper: to identify analysis techniques which may be
leveraged by others in different contexts, without unnecessary technicalities.

About the second point, we note that papers which make Lipschitz-type assumptions about $f$ directly rather than about
$f \circ R_x$ typically restrict the algorithm to use a specific retraction (usually, the potentially expensive exponential map
which computes geodesics). These Riemannian Lipschitz assumptions typically rely on Riemannian distances, parallel
transports along minimizing geodesics, and Riemannian logarithms. The latter are only continuously defined up to the
injectivity radius at each point: this radius limitation (which readily appears in the recent work by Sun et al.) should be
compared to the role of $b$ here. In contrast, formulating assumptions as we do is straightforward and offers flexibility in
choosing the retraction. About assessing whether the regularity assumptions hold, we show that for compact manifolds
and smooth $f$ they always do, for any $b$: this covers a large number of applications. For Euclidean manifolds with
$R_x(s) = x + s$, we recover the classical assumptions exactly. In the general case, we show how to handle any retraction
by bounding its acceleration, which brings further flexibility.

Finally, for the last point, we agree that the need to know the regularity parameters $L, \rho, b$ to run the algorithm is a
downside. We explicitly state this early in the paper, and the same issue also affects Jin et al.'s and Sun et al.'s works.
However, that certain regularity parameters must be known seems inevitable, in particular for the Hessian's $\rho$. Indeed,
the main theorems make statements about the spectrum of the Hessian, yet the algorithm is not allowed to query the
Hessian. If anything, it is remarkable that so little prior knowledge is sufficient. This being said, for structured problems,
the parameters can be determined. For example, in PCA, we seek to maximize $f(x) = x^\top A x$ with some symmetric
matrix $A$ and $x$ living on the unit sphere $S^{n-1} = \{x \in \mathbb{R}^n : x^\top x = 1\}$. A bit of calculus (which we can add to
the appendix) shows that, pullbacked through the classical retraction $R_x(s) = \frac{x+s}{\|x+s\|}$, this cost function satisfies the
regularity conditions with $L = 2.5\|A\|_{\mathrm{op}}$, $\rho = 9\|A\|_{\mathrm{op}}$ and $b = +\infty$. Of course, $\|A\|_{\mathrm{op}}$ can be upper-bounded by the
subordinate norm $\|A\|_1$, which is trivial to compute from $A$. The retraction is second order, hence Assumption 4 holds
with $\beta = 0$. This is sufficient knowledge to run the algorithm: all other parameters are expressed as functions of these
and a user-selected $\varepsilon$.

We thoroughly revised the paper and fine details of the statements and proofs to make it a smoother read. In so doing,
we further clarified technical points that are already important in the Euclidean case but are not discussed by Jin et al.

We hope the reviewers may re-assess our paper in consideration of these points.

[Meta-Review · NeurIPS 2019]

The reviewers initially pointed out that the analysis derived in the paper is rather simple. After discussion, they also agreed that this simplicity is also a strength as the results are much easier to grasp than prior/concurrent work. This could make the analysis easier to extend and re-usable to solve similar problems. Therefore, this seems like a valuable contribution for the community.